# Foliar Spraying with Potassium Bicarbonate Reduces the Negative Impact of Drought Stress on Sweet Basil (*Ocimum basilicum* L.)

**DOI:** 10.3390/plants11131716

**Published:** 2022-06-28

**Authors:** Natalija Burbulis, Aušra Blinstrubienė, Aldona Baltušnikienė, Justina Deveikytė

**Affiliations:** 1Department of Plant Biology and Food Sciences, Vytautas Magnus University, Donelaicio Str. 58, 44248 Kaunas, Lithuania; natalija.burbulis@vdu.lt (N.B.); aldona.baltusnikiene@lsmuni.lt (A.B.); justina.deveikyte@vdu.lt (J.D.); 2Department of Biochemistry, Faculty of Medicine, Lithuanian University of Health Sciences, Mickeviciaus Str. 9, 44307 Kaunas, Lithuania

**Keywords:** antioxidant activity, abiotic stress, chlorophyll, weight, phenolic

## Abstract

In recent years, special attention has been paid to climate change assessment and research into our changing environment. The greatest economic losses worldwide are due to the negative effects of drought stress and extreme temperature on the plants’ morphological, physiological, and biochemical properties which limit crop growth and productivity. Sweet basil (*Ocimum basilicum* L.) is one of the most popular plants widely grown around the world as a spice, as well a medicinal, aromatic plant. The seedlings with 5–6 true leaves were divided into two groups, and one group of seedlings was sprayed with 10 mM potassium bicarbonate (KHCO_3_). Three days after potassium bicarbonate treatment, half of the plants from each group were subjected to a water deficit (drought stress), and the other half were grown under stress-free conditions (well-watered). The present study aimed to evaluate the effect of potassium bicarbonate (KHCO_3_) on morphophysiological parameters, phenolics content and the antioxidant activity of basil under drought conditions. The application of potassium bicarbonate to drought stressed plants significantly increased the chlorophyll content, fresh and dry weight, phenolics content in the two of tested cultivars, and antioxidant activity, determined by DPPH and ABTS methods. Principal component analysis showed that the first factor was highly and positively related to all the investigated parameters. Hierarchical clustering analysis showed that the first cluster was formed by being well-watered, well-watered and sprayed with potassium bicarbonate, and grown under drought conditions and sprayed with potassium bicarbonate basil cultivars, while the second cluster was formed by all the tested cultivars grown under drought conditions.

## 1. Introduction

Sweet basil (*Ocimum basilicum* L.) is one of the most popular plants widely grown around the world as a spice and as a medicinal, aromatic plant. Basil accumulates a complex of biologically active compounds, which determines the therapeutic characteristic of this plant [1]. Basil is very valuable due to its antioxidant [2,3,4], anticancer [5], anticonvulsant [6], and antiphlogistic [7] properties. Suanarunsawat et al. [8] reported that basil leaf extract lowers blood cholesterol and has a positive effect on bile secretion. Laboratory studies have shown that substances with antibiotic activity in basil grass inhibit the harmfulness of pathogenic microorganisms [9].

In recent years, special attention has been paid to climate change assessment and research into our changing environment. A report by the Intergovernmental Panel on Climate Change (IPCC) states that the effects of climate change are being felt on all continents, with global temperatures recently rising by 1.5 °C [10]. Future economic threats are predicted due to increasing droughts and heat outbreaks [11,12]. The greatest economic losses worldwide are due to the negative effect of drought stress and extreme temperature on plants’ morphological, physiological and biochemical properties, limiting crop growth and productivity [13,14,15,16,17,18,19,20,21,22,23].

The negative effect of drought stress on various parameters of plants has been reviewed by Seleiman et al. [14]. Drought has reduced the morphological and physiological traits, reduced leaf water potential and sap movement due to alternation of xylem anatomical features in plants [15]. Zargar et al. [16] states that drought stress has resulted in a decrease in turgor pressure which in turn inhibits cell growth in many aspects. Water loss results in inefficient execution of metabolic processes by influencing the activity of enzymes involved in photosynthesis and thus malfunctioning the photosynthetic apparatus. Enhanced metabolite fluxes cause the formation of free radicals that hinder the development by inducing oxidative stress inside the cell. In studies with apples, Wang et al. [17] found that when leaf water potential is above –1.5 MPa, the stomatal limitation should be the main reason for a drop of photosynthesis; when water potential was below –1.5 MPa, the drop-in photosynthesis activity might be caused by non-stomatal limitation.

Plants have a variety of mechanisms to mitigate the harmful effects of drought. An early response to drought stress helps them to survive [13,18]. Some plants mitigate drought stress by reducing growth and increasing metabolic activities under drought conditions. Farooq et al. [24] state that the main factors that determine physiological plants’ adaptations to the abiotic stress are the lowering of osmotic potential, maintenance of high antioxidant activities, and rearrangement of growth regulators. In the changing climate, the efficiency of agricultural production is primarily determined by the adaptation of plants to stressful situations [25,26]. One way to minimize the harmful effects of stress on plants and optimize nutrient uptake is to spray them with bioactive substances. Although this type of preparation is widely used in agriculture, there is a need for new compounds that regulate plant physiological processes and which have anti-stress properties and are non-toxic to the environment and humans.

Potassium is one of the important macro elements essential for plant growth, development, photosynthesis, metabolic processes, and maintenance of normal osmotic potential [27]. Moreover, potassium helps to stabilize cell membranes, is involved in the transfer of assimilates to roots, and reduces the detrimental effects of abiotic and biotic stresses [28,29]. S. Kanai et al. [30] found that K deficiency can inhibit the activity of aquaporins (water carriers) and reduce water supply to the aboveground part of the plant. Potassium is found in plant cells in the form of K + ions and various salts, including KHCO_3_. Potassium bicarbonate (KHCO_3_) is not detrimental to human health and the environment and is therefore permitted for use in ecological agriculture [31]. Exogenous potassium bicarbonate is used as an effective fungicide [32,33].

In our previous investigations with green foxtail, we found that KHCO_3_ protects photosystem II of green foxtail from drought by significantly increasing the maximum quantum yield, effective quantum yield, and gas exchange parameters [34]. Based on the results of this study, we hypothesized that potassium bicarbonate may increase the drought resistance of sweet basil. The responses of basil to drought stress with the purpose of increasing drought tolerance by various compounds have been studied by certain research groups [35,36,37]; however, to the best of our knowledge, the influence of potassium bicarbonate on the drought response of basil has not been previously reported. The present study aimed to evaluate the effect of potassium bicarbonate (KHCO_3_) on morphophysiological parameters, phenolics content, and the antioxidant activity of basil under drought conditions.

## 2. Results

### 2.1. The Effect of Drought Stress and Potassium Bicarbonate Application on the Chlorophyll a and b Content

The chlorophyll content in well-watered sweet basil varied from 1.93 mg g^−1^ to 2.37 mg g^−1^ depending on the cultivars (Figure 1).

Chlorophyll a content in well-watered plants was not significantly affected by the application of potassium bicarbonate. Under the influence of drought stress, chlorophyll a content decreased from 12% (“Dark Opal” and “Palla Compato”) to 27% (“Cinnamon”) compared to the well-watered plants. The application of potassium bicarbonate to drought stressed plants resulted in increasing chlorophyll a content from 11% (“Aromatico Della Riviera Ligure”) to 22% (“Red Rubin”) compared to the drought stressed plants.

The chlorophyll b content in well-watered sweet basil varied from 0.77 mg g^−1^ to 0.90 mg g^−1^ depending on the cultivars (Figure 2).

The application of potassium bicarbonate slightly increased chlorophyll b content in the well-watered untreated plants; however, differences were not significant in comparison with well-wared untreated plants. Under drought conditions chlorophyll b content in sweet basil plants decreased from 26% (“Red Rubin”) to 34% (“Aromatico Della Riviera Ligure”) compared to the well-watered plants. The application of potassium bicarbonate to drought-stressed plants resulted in the increasing of chlorophyll b content from 22% (“Dark Opal”) to 36% (“Red Rubin”) compared to the drought stressed plants.

### 2.2. The Effect of Drought Stress and Potassium Bicarbonate Application on the Fresh and Dry Weight

The fresh weight of well-watered sweet basil plants varied from 9.30 g to 11.40 g depending on the cultivars (Figure 3).

Spraying with potassium bicarbonate slightly increased the fresh weight of well-watered plants; however, in comparison with well-watered untreated plants, differences were not significant. Under drought conditions, the fresh weight of sweet basil plants decreased from 22% (“Dark Opal”) to 40% (“Red Rubin”) compared to the well-watered plants. The application of potassium bicarbonate to drought stressed plants resulted in the increase of fresh weight from 26% (“Cinnamon”) to 39% (“Red Rubin”) compared to the drought-stressed plants.

The dry weight of well-watered sweet basil plants varied from 1.83 g to 3.17 g depending on the cultivars (Figure 4).

The dry weight of well-watered plants was not significantly affected by application of potassium bicarbonate. Under drought conditions, the dry weight of sweet basil plants decreased from 11% (“Dark Opal”) to 55% (“Palla Compato”) compared to the well-watered plants. The application of potassium bicarbonate to drought stressed plants resulted in increasing dry weight from 34% (“Dark Opal”) to 50% (“Palla Compato”) compared to the drought-stressed plants.

### 2.3. The Effect of Drought Stress and Potassium Bicarbonate Application on the Total Phenolic Content

The total phenolic content in well-watered sweet basil plants varied from 9.56 mg g^−1^ DW to 11.67 mg g^−1^ DW depending on the cultivars (Figure 5).

Significantly increasing total phenolic content under application of potassium bicarbonate on well-watered plants has been observed for the cultivars “Cinnamon”, “Aromatico Della Riviera Ligure” and “Dark Opal”, while total phenolic content in the “Red Rubin” and “Palla Compato” was not significantly affected by application of potassium bicarbonate. As compared to the well-watered plants under drought conditions, the total phenolic content in the “Cinnamon” decreased by 8% (“Dark Opal”) but increased from 7% to 16% in the other tested cultivars. The application of potassium bicarbonate to drought-stressed plants resulted in an increasing of total phenolic content from 14% (“Dark Opal”, “Palla Compato” and “Aromatico Della Riviera Ligure”) to 20% (“Cinnamon”) compared to the drought stressed plants.

### 2.4. The Effect of Drought Stress and Potassium Bicarbonate Application on the Antioxidant Activity

The antioxidant activity measured by the DPPH method from well-watered basil leaf extract varied from 13.67% to 17.19% depending on the cultivars (Figure 6a). Compared to well-watered plants, the application of potassium bicarbonate resulted in an increasing of antioxidant activity in cultivars “Cinnamon”, “Red Rubin”, “Dark Opal” and “Aromatico Della Riviera Ligure” leaf extract, while antioxidant activity measured by DPPH method from leaf extract of cultivar “Palla Compato was not significantly affected by spraying with potassium bicarbonate.

Any significant effect of drought stress condition on antioxidant activity measured by the DPPH method has been obtained for cultivar “Dark Opal”, while antioxidant activity in other cultivars decreased from 9.8% (“Red Rubin”) to 16.4% (“Aromatico Della Riviera Ligure”) in comparison with well-watered plants. The application of potassium bicarbonate on plants grown under drought conditions did not have any significant effects on the antioxidant activity measured by DPPH methods in leaf extracts of cultivar “Red Rubin”, but significantly increased this parameter in other basil cultivars tested.

The antioxidant activity measured by ABTS method from well-watered basil leaf extracts varied from 8.78% to 12.41% depending on the cultivars (Figure 6b). The significant effect of potassium bicarbonate on the antioxidant activity measured by the ABTS method on cultivars “Cinnamon” and “Palla Compato” has not been obtained, while in cultivars “Red Rubin”, “Dark Opal” and “Aromatico Della Riviera Ligure” leaf extracts this parameter increased by 29.6%, 19.4% and 12.7%, respectively. Compared to well-watered plants, antioxidant activity measured by the ABTS method significantly increased in the cultivar “Aromatico Della Riviera Ligure”. The application of potassium bicarbonate significantly increased antioxidant activity measured by ABTS method in cultivars “Cinnamon”, “Red Rubin” and “Aromatico Della Riviera Ligure” plants grown under drought conditions.

### 2.5. Principal Component and Hierarchical Clustering Analysis

The first two components (PCs) were associated with eigenvalues higher than one and explained 52.76% and 30.68% of the total variance (Figure 7). The first factor (PC1) was highly and positively related to all investigated parameters. Figure 7 demonstrated that all treatments and basil cultivars were well separated in the PCA map. The principal component analysis shows that the highest fresh weight, dry weight, and chlorophyll a content were associated with cultivar “Red Rubin” grown under well-watered and drought conditions and sprayed with potassium bicarbonate, while the chlorophyll b content was associated with cultivars “Dark Opal” and “Aromatico Della Riviera Ligure” grown under well-watered conditions, and the total phenolic content and antioxidant activity were associated with cultivars “Cinnamon” and “Aromatico Della Riviera Ligure” grown under drought conditions and sprayed with potassium bicarbonate.

Based on hierarchical clustering analysis, basil cultivars were clustered into two clusters: C1 and C2 (Figure 8). The first cluster was formed by well-watered, well-watered and sprayed with potassium bicarbonate, and grown under drought conditions and sprayed with potassium bicarbonate basil cultivars. The second cluster was formed by all tested cultivars grown under drought conditions.

## 3. Discussion

Drought stress negatively affects plant growth and development with substantial reductions in the crop growth rate and chlorophyll contents, as well as the inhibition of photosynthesis and biomass accumulation [19,22,38]. Drought reduced the chlorophyll content and physiological performance of the plants because chlorophyll helps to capture better light and a higher amount of light. There is also a higher possibility of photosynthetic rate because of the conversation of light energy into chemical energy. However, under drought, the light energy required to change chemicals is comparatively lower due to lower chlorophyll concentration [39]. In our study, the chlorophyll a content under the drought condition significantly decreased, especially in the cultivar “Cinnamon”, while the highest decrease in chlorophyll b content was obtained in the cultivar “Aromatico Della Riviera Ligure”. Chlorophyll a and b decreased under drought conditions obtained in our study and are in agreement with other research groups’ statements about the strong negative effect of drought stress on chlorophyll a and b content in sweet basil, such as those reported by Damalas [36], Zulfiqar et al. [37], Kordi et al. [35], and Al-Hugail et al. [40]. Damalas [36] supposed that such a reduction in chlorophyll may be explained by an imbalance of protein complexes and increased activity of chlorophyll-degrading enzymes and chlorophyllase, while according to Schütz and Fangmeir [41], chlorophyll decreasing is more related to the increase in reactive oxygen species (ROS) production. Moreover, under drought, stress photosynthesis is strongly inhibited by stomatal closure and reduces enzyme activity such as Rubisco [42,43,44]. The application of potassium bicarbonate to drought-stressed plants significantly increased chlorophyll content compared to the drought-stressed plants through the protection of a photosynthetic apparatus and partial recovery of chlorophylls content. The highest positive effect of potassium bicarbonate has been determined for the cultivar “Red Rubin”.

It has been documented that potassium regulates equipoise between anions and cations in the cytoplasm, stimulates ATPase activity, and regulates stoma opening and closing [45,46,47]. Xu et al. [48] found that exogenously applied potassium significantly increased the biomass of apple dwarf rootstock seedlings. A reduction in fresh and dry weight in basil under drought conditions has been reported by Kordi et al. [35] and Barickman et al. [49]. In our study, drought stress significantly decreased the fresh and dry weight of sweet basil in comparison with well-watered plants. Wei et al. [50] reported that the application of K_2_CO_3_ significantly increased shoot and root masses of winter wheat under drought stress in their studies. In our study, the application of potassium bicarbonate to drought-stressed plants resulted in a significant increase in fresh and dry weight compared to the drought-stressed plants.

An increase of total phenolic content under drought stress was observed in sweet basil [35], *Achillea pachycephala* Rech.f. [51], and *Amaranthus tricolor* L. [52]. In contrast, any significant effect of drought stress on phenolic content in tested cultivars has not been determined in our study. The application of potassium bicarbonate significantly increased total phenolic content in the cultivar “Cinnamon” grown under well-watering conditions, as well as in cultivars “Cinnamon” and “Aromatico Della Riviera Ligure” grown under drought stress.

Abiotic stress especially produces the ROS when this plant is exposed to the stress condition and produces antioxidants, flavonoids, and secondary metabolites for protecting the plant for detoxifying ROS and protecting the abnormal condition (i.e., stress) and protein and amino acid stabilization [53,54]. Potassium application had a significant effect on the DPPH radical scavenging capacity of basil leaves in Nguyen et al. [55] studies, but the positive effect was strongly affected by the cultivars. In our study, a significant effect of potassium bicarbonate on DPPH radical scavenging capacity in well-watered plants has not been obtained. On the other hand, potassium bicarbonate significantly increased the antioxidant activity measured by the ABTS method in the well-watered basil cultivars “Red Rubin”, “Dark Opal”, and “Aromatico Della Riviera Ligure”. Moreover, the application of potassium bicarbonate significantly increased the DPPH radical scavenging capacity in basil plants grown under drought conditions. Antioxidant activity measured by ABTS method significantly increased in the cultivars “Cinnamon”, “Red Rubin”, and “Aromatico Della Riviera Ligure” grown under drought stress. Several groups reported a high correlation between total phenolic content and antioxidant activity [56,57,58,59,60]. In our study, free radical scavenging capacity increased with increasing phenols concentration in extracts. A high correlation was found between antioxidant activity data from DPPH (r = 0.99) and ABTS (r = 0.98) assays and total phenolic contents of basil extracts made from different well-watered plants and from basil cultivars with potassium bicarbonate application—r = 0.86 and r = 0.84, respectively.

## 4. Materials and Methods

### 4.1. Experiment Conditions

This investigation was performed with the sweet basil cultivars “Cinnamon”, “Red Rubin”, “Dark Opal”, “Palla Compato”, and “Aromatico Della Riviera Ligure”. The seeds were placed in plastic pots 12 cm in diameter and 10 cm in height and grown in a greenhouse with a 16:8 h photoperiod, 25/22 °C (day/night) temperature, and 150 µmol m^−2^ s^−1^ light density. The pots were filled with a substrate which was composed of peat and sand at 3:1 volume ratio. At the cotyledon stage of seedlings, they were thinned to five per pot. The seedlings with the 5–6 true leaves were divided into 2 groups, and one group of seedlings were sprayed with 10 mM potassium bicarbonate (KHCO_3_). Three days after potassium bicarbonate treatment half of the plants from each group were subjected to a water deficit (drought stress), and the other half were grown under stress-free conditions (well-watered). The samples for testing were collected after seven weeks of germination.

### 4.2. Determination of Chlorophyll Content

Chlorophyll a and b content were determined according to Sims, Gamon [61]. A total of 0.2 g of fresh well-developed leaves was homogenized with acetone (80% *v*/*v*) and then centrifuged at 3000 rpm. The supernatant was used to measure of absorbance at 663 and 647 nm using spectrophotometer *Spectro UV-VIS Dual beam* (Labomed Inc., Los Angeles, CA, USA).

### 4.3. Determination of Fresh and Dry Weight

At the end of the experiment, all plants in each pot were cut at the soil surface, and shoot fresh weight was recorded by a digital scale. Shoots were dried in a thermostat under 55 °C temperature and their dry weight was recorded by digital scale.

### 4.4. Determination of Phenolic Compound Contents, DPPH and ABTS Radical Scavenging Capacity

The phenolic compound contents DPPH and ABTS radical scavenging capacity were determined in 96% ethanolic extracts prepared by homogenizing 0.3 g plant material in 10 mL of 96% ethanol solution. The homogenates were centrifuged at 4500× *g* for 30 min, and the supernatant was used for further analysis. The Folin–Ciocalteu method was used for the determination of the total phenolic compound content. To a test tube containing 100 µL of plant extract 300 µL of 0.2 M Folin–Ciocalteu, reagent was added and incubated for 10 min at room temperature in darkness. After that, 5 mL of 7.5% Na_2_CO_3_ solution was added and the absorbance of the reaction mixture was measured after 30 min of incubation (room temperature, darkness) at 765 nm using the spectrophotometer *Spectro UV-VIS Dual beam* (Labomed Inc., Los Angeles, CA, USA). The estimation of TPC was performed using a calibration curve with gallic acid solutions as standards. Samples were analyzed in triplicate. Results were expressed as mg gallic acid equivalent (GAE) per gram dry weight.

The total antioxidant activity of basil extracts was measured using DPPH (1,1-diphenyl-2-picrylhydrazyl) and ABTS+ (2,20-azino-bis (3-ethylbenzothiazoline-6-sulfonic acid) radical scavenging activity assays as described by Yim, Nam [62]. For analysis, 96% ethanolic extracts from ground basil leaves were used. The antioxidant activity was expressed as mg Trolox equivalents (TE) per gram dry weight. Before analysis, the 60 µM DPPH stock solution was diluted with ethanol to give an absorbance of 0.8 ± 0.03 at 517 nm. For analysis, the 10 µL of basil extract and 3 mL of diluted DPPH solution were added to the tube and shaken vigorously, followed by incubation at room temperature for 30 min in the dark. 

The decrease in absorbance in the presence of DPPH was measured at 517 nm using a spectrophotometer (UVS-2800, Labomed Inc., Los Angeles, CA, USA). The radical scavenging capacity was expressed as a % of the inhibition of DPPH· radicals using the following equation: DPPH radical scavenging capacity (%) = (1 − (A1/A0)) × 100. Where: A0 is the absorbance of the DPPH itself; A1 is the absorbance of sample. The ABTS radical scavenging activity of the basil extracts was measured by the ABTS cation decolorization assay. The ABTS radical cation (ABTS+·) was produced by a reaction of 2 mM ABTS stock solution with 0.0095 g potassium persulfate, which was allowed to stand in the dark at room temperature for 16 h before use. Before analysis, the ABTS + solution was diluted with ethanol to give an absorbance of 0.8 ± 0.03 at 734 nm. For analysis, 20 µL of ethanolic basil extract was allowed to react with 3 mL of the ABTS+ solution for 60 min in the dark, after the absorbance of the mixture was measured at 734 nm. The radical scavenging capacity was expressed as the % of the inhibition of ABTS radicals using the following equation: ABTS radical scavenging capacity (%) = (1 − (A1/A0)) × 100, where: A0 is the absorbance of the ABTS itself; A1 is the absorbance of sample.

### 4.5. Statistical Analysis

Experiments were arranged with a complete randomization, and the assay was performed in triplicate. The data analysis was computed using the software package TIBCO Statistica, version 10 (TIBCO Software, Palo Alto, CA, USA). The statistical difference (*p* < 0.05) among the means was analyzed by Tukey’s post hoc test. The mean values of the chlorophyll a, chlorophyll b, fresh weight, dry weight, phenolic compound contents, DPPH and ABTS radical scavenging capacity, and standard error (SE) were calculated on the basis of the number of independent replicates. The effect of factors (drought stress and potassium bicarbonate) and their interaction with investigated variables were studied by two-way ANOVA. The Tukey’s honestly significant difference (HSD) test was carried out with the significance level of *p* < 0.05. The principal component and hierarchical clustering analysis were performed to evaluate the relationships between the drought stress and application of potassium bicarbonate on the evaluated parameters with XLSTAT software version 2021.3.1 (Addinsoft, Paris, France).

## 5. Conclusions

In the present study, the effects of exogenously applied potassium bicarbonate on the morphophysiological parameters, phenolic content, and the antioxidant activity of sweet basil under drought conditions were investigated. The obtained results showed that the application of potassium bicarbonate to drought-stressed plants resulted in significant increases in chlorophyll content, fresh and dry weight, total phenolic content in two of the tested cultivars, and antioxidant activity. The application of potassium bicarbonate to drought-stressed plants significantly increased chlorophyll content compared to the drought-stressed plants through the protection of a photosynthetic apparatus and partial recovery of chlorophyll content. Moreover, the application of potassium bicarbonate to drought-stressed plants resulted in a significant increase in fresh, dry weight, and total phenolic content compared to the drought-stressed plants. Based on our results, foliar spraying with potassium bicarbonate could be an effective method to mitigate the negative effects of drought stress on sweet basil.

## Figures and Tables

**Figure 1 plants-11-01716-f001:**
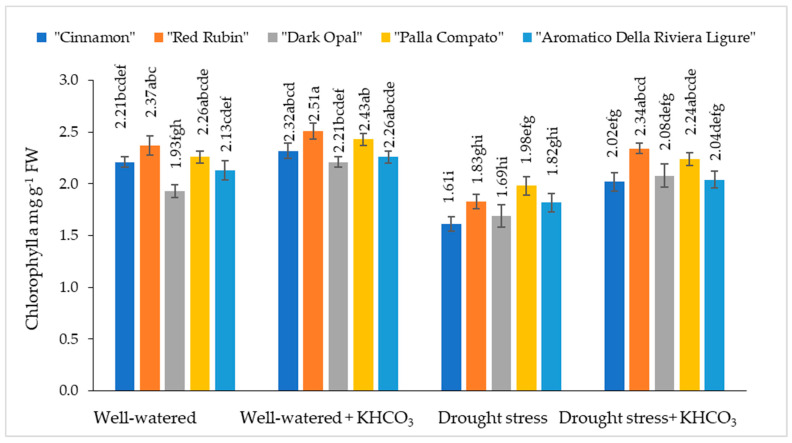
The effect of drought stress and potassium bicarbonate application on the chlorophyll a content in sweet basil plants under well-watered and drought stress conditions. Treatments were performed in triplicate; statistical significance was assayed by two-way ANOVA, followed by a Tukey post hoc test (*p* < 0.05). Data are expressed as mean ± standard error. Means not sharing a common letter are significantly different (*p* ˂ 0.05).

**Figure 2 plants-11-01716-f002:**
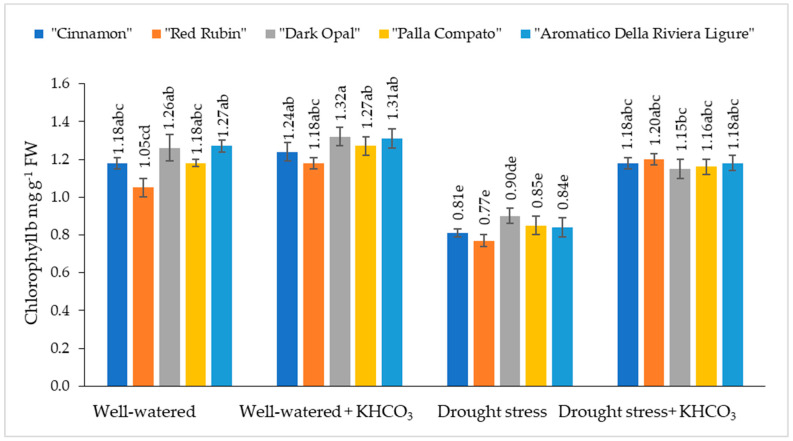
The effect of drought stress and potassium bicarbonate application on chlorophyll b content in sweet basil plants under well-watered and drought stress conditions. Treatments were performed in triplicate; statistical significance was assayed by two-way ANOVA, followed by a Tukey post hoc test (*p* < 0.05). Data are expressed as mean ± standard error. Means not sharing a common letter are significantly different (*p* ˂ 0.05).

**Figure 3 plants-11-01716-f003:**
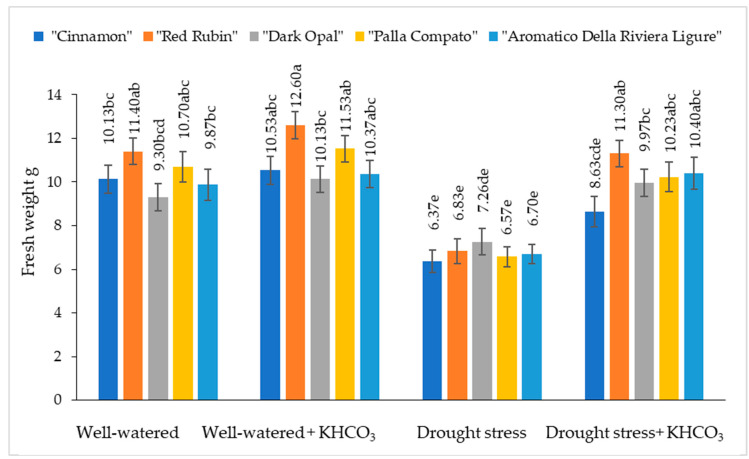
The effect of drought stress and potassium bicarbonate application on the fresh weight of sweet basil plants under well-watered and drought stress conditions. Treatments were performed in triplicate; statistical significance was assayed by two-way ANOVA, followed by a Tukey post hoc test (*p* < 0.05). Data are expressed as mean ± standard error. Means not sharing a common letter are significantly different (*p* ˂ 0.05).

**Figure 4 plants-11-01716-f004:**
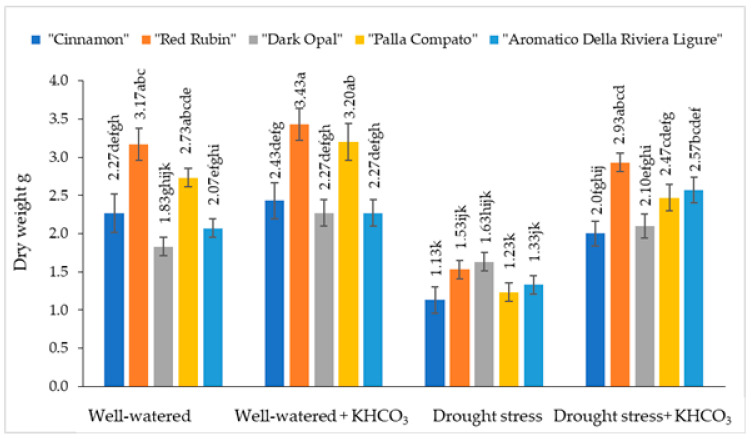
The effect of drought stress and potassium bicarbonate application on the dry weight of sweet basil plants under well-watered and drought stress conditions. Treatments were performed in triplicate; statistical significance was assayed by two-way ANOVA, followed by a Tukey post hoc test (*p* < 0.05). Data are expressed as mean ± standard error. Means not sharing a common letter are significantly different (*p* ˂ 0.05).

**Figure 5 plants-11-01716-f005:**
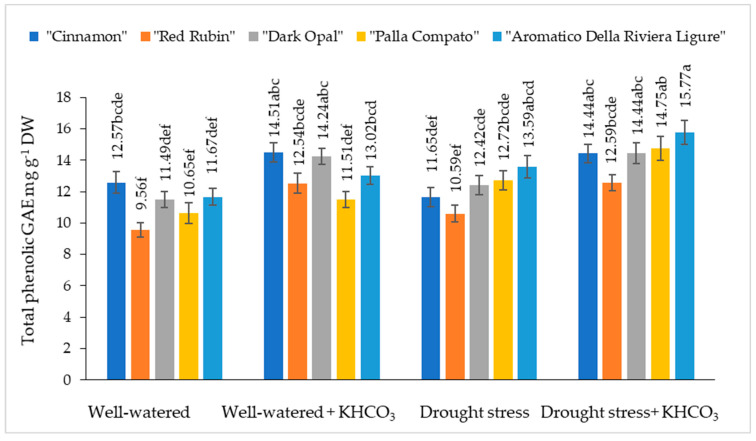
The effect of drought stress and potassium bicarbonate application on total phenolic content in sweet basil plants under well-watered and drought stress conditions. Treatments were performed in triplicate; statistical significance was assayed by two-way ANOVA, followed by a Tukey post hoc test (*p* < 0.05). Data are expressed as mean ± standard error. Means not sharing a common letter are significantly different (*p* ˂ 0.05).

**Figure 6 plants-11-01716-f006:**
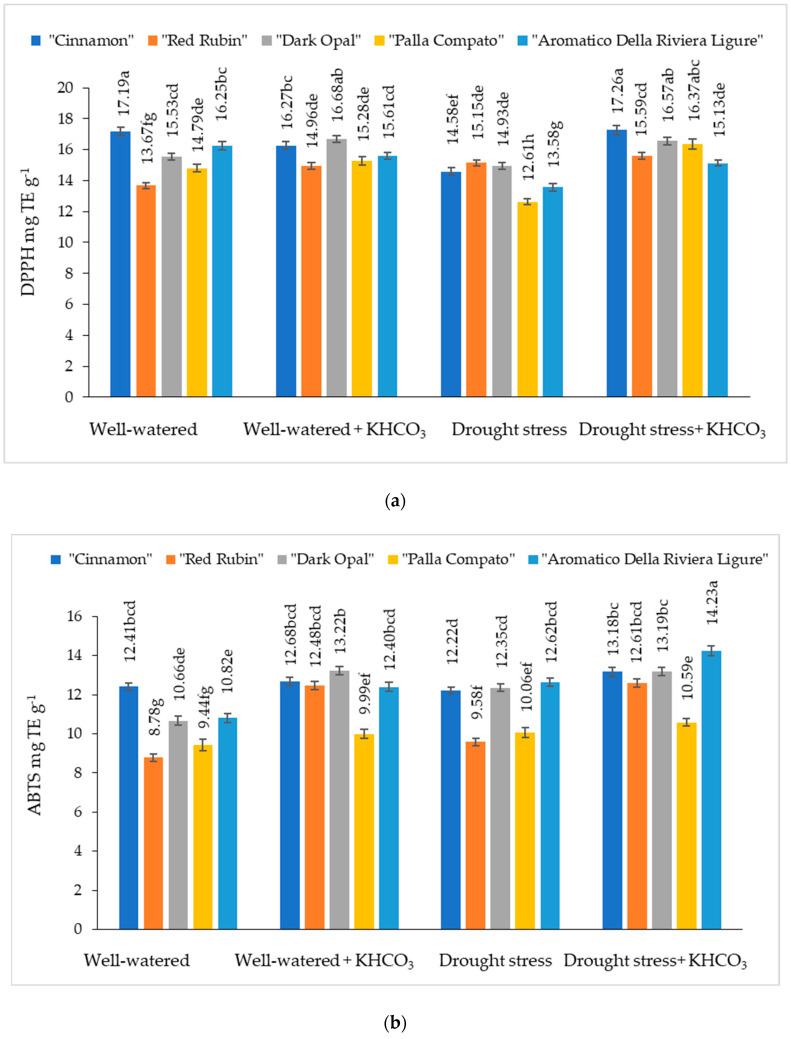
The effect of drought stress and potassium bicarbonate application on antioxidant activity of basil leaf extracts determined by DPPH (**a**) and ABTS (**b**) assays. Treatments were performed in triplicate; statistical significance was assayed by two-way ANOVA, followed by a Tukey post hoc test (*p* < 0.05). Data are expressed as mean ± standard error. Means not sharing a common letter are significantly different (*p* ˂ 0.05).

**Figure 7 plants-11-01716-f007:**
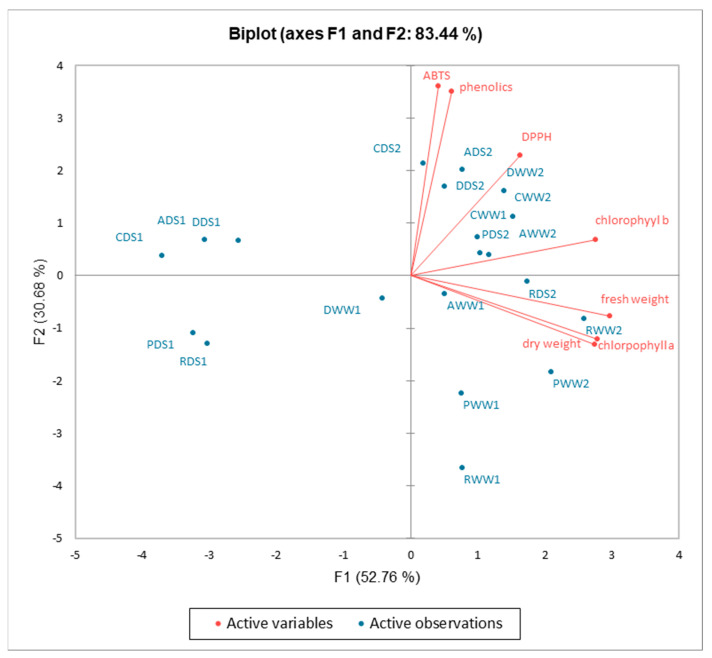
A principal component analysis for chlorophylls content, fresh and dry weight, total phenolics content, and antioxidant activity of basil leaf extracts influenced by drought stress and potassium bicarbonate application. CWW1–“Cinnamon” well-watered; CWW2–“Cinnamon” well-watered and sprayed with potassium bicarbonate; CDS1–“Cinnamon” drought stress; CDS2–“Cinnamon” drought stress and sprayed with potassium bicarbonate; RWW1–“Red Rubin” well-watered; RWW2–“Red Rubin” well-watered and sprayed with potassium bicarbonate; RDS1–“Red Rubin” well-watered; RDS2–“Red Rubin” well-watered and sprayed with potassium bicarbonate; DWW1–“Dark Opal” well-watered; DWW2–“Dark Opal” well-watered and sprayed with potassium bicarbonate; DDS1–“Dark Opal” drought stress; DDS2–“Dark Opal” drought stress and sprayed with potassium bicarbonate; PWW1–“Palla Compato” well-watered; PWW2–“Palla Compato” well-watered and sprayed with potassium bicarbonate; PDS1–“Palla Compato” drought stress; PDS2–“Palla Compato” drought stress and sprayed with potassium bicarbonate; AWW1–“Aromatico Della Riviera Ligure” well-watered; AWW2–“Aromatico Della Riviera Ligure” well-watered and sprayed with potassium bicarbonate; ADS1–“Aromatico Della Riviera Ligure” well-watered; ADS2–“Aromatico Della Riviera Ligure” well-watered and sprayed with potassium bicarbonate.

**Figure 8 plants-11-01716-f008:**
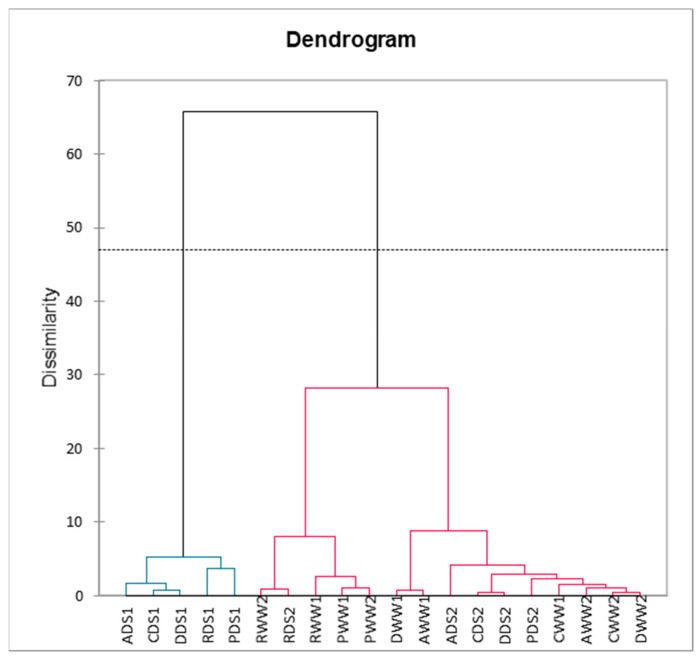
A hierarchical clustering analysis for chlorophylls content, fresh and dry weight, total phenolics content, and antioxidant activity of basil leaf extracts influenced by drought stress and potassium bicarbonate application. CWW1—“Cinnamon” well-watered; CWW2—“Cinnamon” well-watered and sprayed with potassium bicarbonate; CDS1—“Cinnamon” drought stress; CDS2—“Cinnamon” drought stress and sprayed with potassium bicarbonate; RWW1—“Red Rubin” well-watered; RWW2—“Red Rubin” well-watered and sprayed with potassium bicarbonate; RDS1—“Red Rubin” well-watered; RDS2—“Red Rubin” well-watered and sprayed with potassium bicarbonate; DWW1—“Dark Opal” well-watered; DWW2—“Dark Opal” well-watered and sprayed with potassium bicarbonate; DDS1—“Dark Opal” drought stress; DDS2—“Dark Opal” drought stress and sprayed with potassium bicarbonate; PWW1—“Palla Compato” well-watered; PWW2—“Palla Compato” well-watered and sprayed with potassium bicarbonate; PDS1—“Palla Compato” drought stress; PDS2—“Palla Compato” drought stress and sprayed with potassium bicarbonate; AWW1—“Aromatico Della Riviera Ligure” well-watered; AWW2—“Aromatico Della Riviera Ligure” well-watered and sprayed with potassium bicarbonate; ADS1—“Aromatico Della Riviera Ligure” well-watered; ADS2—“Aromatico Della Riviera Ligure” well-watered and sprayed with potassium bicarbonate.

## Data Availability

Not Applicable.

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
