# Peer review of "Foliar Spraying with Potassium Bicarbonate Reduces the Negative Impact of Drought Stress on Sweet Basil (Ocimum basilicum L.)"

_plants, 2022, doi:10.3390/plants11131716_

Round 1
Reviewer 1 Report
I have reviewed the manuscript “Foliar spraying with potassium bicarbonate reduces the negative impact of drought stress on sweet basil”. My comments and suggestions are as follow:
Please start your abstract with the negative impact of drought stress on plants species and then you can discuss about the sweet basil.
Also write scientific name of sweet basil in the title.
The application of potassium bicarbonate to drought stressed plants significantly increased the chlorophyll content, fresh and dry weight, phenolics content in the two of tested cultivars, and antioxidant activity” Please specify the antioxidants?
Please note that keywords should not be repeated which already present in the title.
“Plants have a variety of mechanisms to mitigate the harmful effects of drought. An early response to drought stress helps them survive”. Please add the following citations: ( Azeem et al., 2022).
Please write some more details about the effects of drought stress on plants?
“Drought stress negatively affect plant growth and development with substantial reductions in the crop growth rate and chlorophyll contents, and the inhibition of photosynthesis and biomass accumulation” (Ghafar et al., 2021). Please add this.
Other portion looks fine.
Good Luck.
References
Azeem, M., Haider, M.Z., Javed, S., Saleem, M.H., Alatawi, A., 2022. Drought Stress Amelioration in Maize (Zea mays L.) by Inoculation of Bacillus spp. Strains under Sterile Soil Conditions. Agriculture 12, 50.
Ghafar, M.A., Akram, N.A., Saleem, M.H., Wang, J., Wijaya, L., Alyemeni, M.N., 2021. Ecotypic Morphological and Physio-Biochemical Responses of Two Differentially Adapted Forage Grasses, Cenchrus ciliaris L. and Cyperus arenarius Retz. to Drought Stress. Sustainability 13, 8069.
Author Response
Dear Reviewer,
Thank you very much for carefully review and valuable suggestions.
Point 1: Please start your abstract with the negative impact of drought stress on plants species and then you can discuss about the sweet basil.
Response 1: Correction “Abstract: In recent years, special attention has been paid to climate change assessment and research into the changing environment. The greatest economic losses worldwide are due to the negative effects of drought stress and extreme temperature on the plants’ morphological, physiological, and biochemical properties which limit crop growth and productivity….”
Point 2: Also write scientific name of sweet basil in the title.
Response 2: Correction “Foliar spraying with potassium bicarbonate reduces the negative impact of drought stress on sweet basil (Osimum basilicum L.)”
Point 3: The application of potassium bicarbonate to drought stressed plants significantly increased the chlorophyll content, fresh and dry weight, phenolics content in the two of tested cultivars, and antioxidant activity” Please specify the antioxidants?
Response 3: Correction “The application of potassium bicarbonate to drought stressed plants significantly increased the chlorophyll content, fresh and dry weight, phenolics content in the two of tested cultivars, and antioxidant activity, determined by DPPH and ABTS methods.” (
Point 4: Please note that keywords should not be repeated which already present in the title.
Response 4: Correction “Keywords: antioxidant activity; abiotic stress; chlorophyll; weight; phenolic”
Point 5: “Plants have a variety of mechanisms to mitigate the harmful effects of drought. An early response to drought stress helps them survive”. Please add the following citations:( Azeem et al., 2022).
Response 5: Mentioned citation was added.
Point 6: Please write some more details about the effects of drought stress on plants?
Response 6: Sentence “The negative effect of drought stress on various parameters of plants has been reviewed by Seleiman et al. [14].” Is inserted into the introduction.
Point 7: “Drought stress negatively affect plant growth and development with substantial reductions in the crop growth rate and chlorophyll contents, and the inhibition of photosynthesis and biomass accumulation” (Ghafar et al., 2021). Please add this.
Response 7: Suggested reference is added.

Reviewer 2 Report
General comments
I have read the manuscript (plants-1734870). Entitle: Foliar spraying with potassium bicarbonate reduces the negative impact of drought stress on sweet basil written by Natalija Burbulis et. al., for publication of plants MDPI. In this study, the author investigates the evaluate the effect of potassium bicarbonate (KHCO3) on morphophysiological and biochemical traits of the basil under drought conditions. The application of potassium bicarbonate to drought-stressed plants significantly increased the chlorophyll content, fresh and dry weight, phenolics content in the two of tested cultivars, and antioxidant activity.
The overall research is well conducted, and research is obvious application potential for the readers because that helps to the overall effect of the drought stress and different traits of the plant and their relationship with the other biochemical traits. In this sense, the manuscript is much valuable. However, I found some points, especially the flow of the text and lack of potential references, and lack of connection of story in different paragraphs, especially in the introduction and discussion sections. The author should provide enough examples and their interpretation of different traits of physiological and biochemical responses by the latest and appropriate references, some of which I mentioned below. Overall after I evaluate and request the author for this manuscript as a “MAJOR REVISION”.
Major suggestions
1) Introduction: The introduction is well starting with the background of sweet basil and its medicinal and its antioxidant properties. However, throughout the introduction, there is not well mention of the overall negative effect due to the drought stress, which is very important because drought studies the main theme of this research. Please read and mentioned this as a reference. Entitle: Entitle “Response of drought stress in prunus sargentii and larix kaempferii ...https://doi.org/10.1016/j.foreco.2020.118099” Please mentioned that “drought reduced the morphological and physiological traits, reduce the leaf water potential and sap movement due to alternation of xylem anatomical features in the plants”.
2) Hypothesis and objectives of the study: Author should rephrase the text especially the last section of the introduction Line no. 76. In the last paragraph, the author should present the hypothesis of the study and at the same time, author should connect the hypothesis of the study with the research objectives. Please revise the Ln 76-84 more logically. The hypothesis of the study is an important thing, and it gives another strength to the introduction. The hypothesis should be very clear in the introduction sections because, without appropriate literature, questions, or hypotheses in the introduction section the entire text will be unclear. The author should give special attention and the sequential presentation of the content in the introduction with presenting the hypothesis of the study.
3) Discussion: Author should Improve the subsection of discussion 2.4 (line no. 197) more logically with clearer potential references because the main theme of antioxidant and secondary metabolites under drought stress conditions and release the ROS (why ROS is emerging in stress conditions?). Refer to these two articles for better clarify (1) https://doi.org/10.1038/s41598-019-55889 (2) https://doi.org/10.1016/j.scitotenv.2021.146466 and mention somewhere in that paragraph “abiotic stress especially environmental stress (I.e. drought) plant produces the ROS when these plant exposed to the stress condition and plant produce antioxidant, flavonoids, and secondary metabolites play to the role for protecting the plant for detoxifying ROS and protect the plant to protect the abnormal condition (i.e. stress) and protein and amino acid stabilization”.
Other minor suggestions
4) Line no. 107:
Author should be focusing the why chlorophyll is reduced and what is the negative effect due to the reduction of Chl. under drought stress. Reduction of Chl under drought stress condition author can refer to this article as a references “DOI:10.1016/j.scienta.2018.11.021. Author should mention that the “drought reduced the Chlorophyll content and that reduce the physiological performance of the plants because chlorophyll help to capture the better light and higher amount of light due to Chl. then higher possibility of Pn because of conversation of light energy change into the chemical energy, however, under drought the light energy to change chemical is comparatively lower due to lower Chl concentration”.
5) Line no. 779 (M&M section): Author should be more focused on the MM section which is comparatively short and each of the methodologies are not clear. The author should be special care in this section.
6) Line no. 792: Author should clearly mention all the methodology of how author extracts the chlorophyll from the leaf. The author should mention the steps shortly. I do not see this information in the 4.2 section. Or author should refer the literature from which the procedure author followed to extract the chlorophyll content.
7) Line no. 879 (Conclusion): The conclusion for me comes off as repetitive of the abstract or a summary of the results section. I would love to read striking points and take-home messages that will linger in the readers’ minds. What is the novelty, how does the study elucidate some questions in this field, and the contributions the paper may offer to the scientific community?
8) Line no. 897 (Reference): please double-check the citations, their style, spell check, and other grammatical errors. moreover, I request to the authors for revision throughout the manuscript according to the journal rules.
Good Luck!
Author Response
Dear Reviewer,
Thank you very much for carefully review and valuable suggestions.
Point 1: Line previously 307 now 790??: It should be administered after how many days after the application of the spray treatments, the plants for testing were harvested.
Response 1: Plants were harvested after three weeks of the spray treatments, but unfortunately, we don’t know where must we administered that, could you please advise us.
Point 2: Line 311: It is not indicated that chlorophyll was determined in fresh leaves, this can be emphasized for the sake of clarity- given.
Response 2: Correction “A total of 0.2 g of fresh well-developed leaves was homogenized with acetone (80% v/v) and then centrifuged at 3000 rpm.” (Line 978).
Point 3: Line previously 315, now 798: To be given after how many days of the experiment (from sowing or germination) plants were harvested.
Response 3: Correction “The samples for testing were collected after seven weeks of germination.” (Line 975).

Reviewer 3 Report
I maintain that the manuscript entitled „Foliar spraying with potassium bicarbonate reduces the negative impact of drought stress on sweet basil” submitted to Plants journal is well written and the results are presented in a logical and coherent manner.
The paper is adequately organized and the topic is interesting and focuses on the impact assessment of the effect of potassium bicarbonate on morphophysiological parameters, phenol content and antioxidant activity of basil in drought conditions. Supporting crops during drought stress is one of the world's key challenges in terms of climate change. Therefore, I believe that the results of the research carried out deserve to be published in a widely available format.
Despite the improvement of the manuscript, the Authors have not addressed all of my previous comments
Line previously 307 now 790??: It should be administered after how many days after the application of the spray treatments, the plants for testing were harvested
Line 311: It is not indicated that chlorophyll was determined in fresh leaves, this can be emphasized for the sake of clarity - given
Line previously 315, now 798: To be given after how many days of the experiment (from sowing or germination) plants were harvested
Author Response
We obtained reports of 2 Reviewers
Round 2
Reviewer 2 Report
General comments
I have read the manuscript (plants-1734870). Entitle: Foliar spraying with potassium bicarbonate reduces the negative impact of drought stress on sweet basil written by Natalija Burbulis et. al., for publication of plants MDPI. In this study, the author investigates the evaluate the effect of potassium bicarbonate (KHCO3) on morphophysiological and biochemical traits of the basil under drought conditions. The application of potassium bicarbonate to drought-stressed plants significantly increased the chlorophyll content, fresh and dry weight, phenolics content in the two of tested cultivars, and antioxidant activity.
The overall research is well conducted, and research is obvious application potential for the readers because that helps to the overall effect of the drought stress and different traits of the plant and their relationship with the other biochemical traits. In this sense, the manuscript is much valuable. However, I found some points, especially the flow of the text and lack of potential references, and lack of connection of story in different paragraphs, especially in the introduction and discussion sections. The author should provide enough examples and their interpretation of different traits of physiological and biochemical responses by the latest and appropriate references, some of which I mentioned below. Overall after I evaluate and request the author for this manuscript as a “MAJOR REVISION”.
Major suggestions
1) Introduction: The introduction is well starting with the background of sweet basil and its medicinal and its antioxidant properties. However, throughout the introduction, there is not well mention of the overall negative effect due to the drought stress, which is very important because drought studies the main theme of this research. Please read and mentioned this as a reference. Entitle: Entitle “Response of drought stress in prunus sargentii and larix kaempferii ...https://doi.org/10.1016/j.foreco.2020.118099” Please mentioned that “drought reduced the morphological and physiological traits, reduce the leaf water potential and sap movement due to alternation of xylem anatomical features in the plants”.
2) Hypothesis and objectives of the study: Author should rephrase the text especially the last section of the introduction Line no. 76. In the last paragraph, the author should present the hypothesis of the study and at the same time, author should connect the hypothesis of the study with the research objectives. Please revise the Ln 76-84 more logically. The hypothesis of the study is an important thing, and it gives another strength to the introduction. The hypothesis should be very clear in the introduction sections because, without appropriate literature, questions, or hypotheses in the introduction section the entire text will be unclear. The author should give special attention and the sequential presentation of the content in the introduction with presenting the hypothesis of the study.
3) Discussion: Author should Improve the subsection of discussion 2.4 (line no. 197) more logically with clearer potential references because the main theme of antioxidant and secondary metabolites under drought stress conditions and release the ROS (why ROS is emerging in stress conditions?). Refer to these two articles for better clarify (1) https://doi.org/10.1038/s41598-019-55889 (2) https://doi.org/10.1016/j.scitotenv.2021.146466
and mention somewhere in that paragraph “abiotic stress especially environmental stress (I.e. drought) plant produces the ROS when these plant exposed to the stress condition and plant produce antioxidant, flavonoids, and secondary metabolites play to the role for protecting the plant for detoxifying ROS and protect the plant to protect the abnormal condition (i.e. stress) and protein and amino acid stabilization”.
Other minor suggestions
4) Line no. 107:
Author should be focusing the why chlorophyll is reduced and what is the negative effect due to the reduction of Chl. under drought stress. Reduction of Chl under drought stress condition author can refer to this article as a references “DOI:10.1016/j.scienta.2018.11.021. Author should mention that the “drought reduced the Chlorophyll content and that reduce the physiological performance of the plants because chlorophyll help to capture the better light and higher amount of light due to Chl. then higher possibility of Pn because of conversation of light energy change into the chemical energy, however, under drought the light energy to change chemical is comparatively lower due to lower Chl concentration”.
5) Line no. 779 (M&M section): Author should be more focused on the MM section which is comparatively short and each of the methodologies are not clear. The author should be special care in this section.
6) Line no. 792: Author should clearly mention all the methodology of how author extracts the chlorophyll from the leaf. The author should mention the steps shortly. I do not see this information in the 4.2 section. Or author should refer the literature from which the procedure author followed to extract the chlorophyll content.
7) Line no. 879 (Conclusion): The conclusion for me comes off as repetitive of the abstract or a summary of the results section. I would love to read striking points and take-home messages that will linger in the readers’ minds. What is the novelty, how does the study elucidate some questions in this field, and the contributions the paper may offer to the scientific community?
8) Line no. 897 (Reference): please double-check the citations, their style, spell check, and other grammatical errors. moreover, I request to the authors for revision throughout the manuscript according to the journal rules.
Good Luck!
Author Response
Response to Reviewer 2 Comments
Dear Reviewer,
Thank you very much for carefully review and valuable suggestions.
Point 1: The introduction is well starting with the background of sweet basil and its medicinal and its antioxidant properties. However, throughout the introduction, there is not well mention of the overall negative effect due to the drought stress, which is very important because drought studies the main theme of this research. Please read and mentioned this as a reference. Entitle: Entitle “Response of drought stress in prunus sargentii and larix kaempferii...https://doi.org/10.1016/j.foreco.2020.118099” Please mentioned that “drought reduced the morphological and physiological traits, reduce the leaf water potential and sap movement due to alternation of xylem anatomical features in the plants”.
Response 1: Correction “Drought reduced the morphological and physiological traits, reduce the leaf water potential and sap movement due to alternation of xylem anatomical features in the plants.” (Lines 65-67)
Point 2: Author should rephrase the text especially the last section of the introduction Line no. 76. In the last paragraph, the author should present the hypothesis of the study and at the same time, author should connect the hypothesis of the study with the research objectives. Please revise the Ln 76-84 more logically. The hypothesis of the study is an important thing, and it gives another strength to the introduction. The hypothesis should be very clear in the introduction sections because, without appropriate literature, questions, or hypotheses in the introduction section the entire text will be unclear. The author should give special attention and the sequential presentation of the content in the introduction with presenting the hypothesis of the study.
Response 2: Correction “Based on the results of this study we hypothesized that potassium bicarbonate may increase the drought resistance of sweet basil.” (Lines 91-93).
Point 3: Discussion: Author should Improve the subsection of discussion 2.4 (line no. 197) more logically with clearer potential references because the main theme of antioxidant and secondary metabolites under drought stress conditions and release the ROS (why ROS is emerging in stress conditions?). Refer to these two articles for better clarify (1) https://doi.org/10.1038/s41598-019-55889 (2) https://doi.org/10.1016/j.scitotenv.2021.146466 and mention somewhere in that paragraph “abiotic stress especially environmental stress (I.e. drought) plant produces the ROS when these plant exposed to the stress condition and plant produce antioxidant, flavonoids, and secondary metabolites play to the role for protecting the plant for detoxifying ROS and protect the plant to protect the abnormal condition (i.e. stress) and protein and amino acid stabilization”.
Response 3: Correction “Abiotic stress especially environmental stress plant produces the ROS when this plant exposed to the stress condition and plant produce antioxidant, flavonoids, and secondary metabolites play to the role for protecting the plant for detoxifying ROS and protect the plant to protect the abnormal condition (i.e., stress) and protein and amino acid stabilization [52,53].” (Lines 744-748).
Point 4: Line no. 107: Author should be focusing the why chlorophyll is reduced and what is the negative effect due to the reduction of Chl. under drought stress. Reduction of Chl under drought stress condition author can refer to this article as a references “DOI:10.1016/j.scienta.2018.11.021. Author should mention that the “drought reduced the Chlorophyll content and that reduce the physiological performance of the plants because chlorophyll help to capture the better light and higher amount of light due to Chl. then higher possibility of Pn because of conversation of light energy change into the chemical energy, however, under drought the light energy to change chemical is comparatively lower due to lower Chl concentration”.
Response 4: Correction “Drought reduced the morphological and physiological traits, reduce the leaf water potential and sap movement due to alternation of xylem anatomical features in the plants [15].” (Lines 65-67).
Point 5: Line no. 779 (M&M section): Author should be more focused on the MM section which is comparatively short and each of the methodologies are not clear. The author should be special care in this section.
Response 5: We are very sorry, but it seems to us that there is some misunderstanding in this place, because in the first version of the manuscript, which was submitted on April 20, the section Materials and Methods starts from Line 296, in the second version – from Line 887 and in the present version – from Line 998.
Point 6: Line no. 792: Author should clearly mention all the methodology of how author extracts the chlorophyll from the leaf. The author should mention the steps shortly. I do not see this information in the 4.2 section. Or author should refer the literature from which the procedure author followed to extract the chlorophyll content.
Response 6: Section 4.2 “Chlorophyll a and b content were determined according to Sims, Gamon [56]. A total of 0.2 g of fresh well-developed leaves was homogenized with acetone (80% v/v) and then centrifuged at 3000 rpm. The supernatant was used to measure of absorbance at 663 and 647 nm using spectrophotometer Spectro UV-VIS Dual beam (Labomed Inc, USA).”
Point 7: Line no. 879 (Conclusion): The conclusion for me comes off as repetitive of the abstract or a summary of the results section. I would love to read striking points and take-home messages that will linger in the readers’ minds. What is the novelty, how does the study elucidate some questions in this field, and the contributions the paper may offer to the scientific community?
Response 7: Corrections “Conclusions
In present study, the effects of exogenously applied potassium bicarbonate on the morphophysiological parameters, phenolic content and the antioxidant activity of sweet basil under drought conditions were investigated. The obtained results showed that the application of potassium bicarbonate to drought-stressed plants resulted in significant increases in chlorophyll content, fresh and dry weight, total phenolic content in the two of tested cultivars, and antioxidant activity. The application of potassium bicarbonate to drought-stressed plants significantly increased chlorophyll content compared to the drought-stressed plants through protection of photosynthetic apparatus and partial recovery of chlorophylls content. Moreover, the application of potassium bicarbonate to drought-stressed plants resulted in a significant increase in fresh, dry weight and total phenolic content compared to the drought-stressed plants. Based on our results, foliar spraying with potassium bicarbonate could be an effective method to mitigate the negative effects of drought stress on sweet basil.”
Point 8: Line no. 897 (Reference): please double-check the citations, their style, spell check, and other grammatical errors. moreover, I request to the authors for revision throughout the manuscript according to the journal rules.
Response 8: The English language was corrected by experts from the MDPI English Editing service.

This manuscript is a resubmission of an earlier submission. The following is a list of the peer review reports and author responses from that submission.
Round 1
Reviewer 1 Report
Manuscript entitled „Foliar spraying with potassium bicarbonate reduces the negative impact of drought stress on sweet basil” submitted to Plants journal is well written and the results are presented in a logical and coherent manner.
The paper is adequately organized and the topic is interesting and focuses on the impact assessment of the effect of potassium bicarbonate on morphophysiological parameters, phenol content and antioxidant activity of basil in drought conditions. Supporting crops during drought stress is one of the world's key challenges in terms of climate change. Therefore, I believe that the results of the research carried out deserve to be published in a widely available format.
Although the manuscript is well-edited, however small improvements should be introduced that will improve its quality:
Line 307: It should be administered after how many days after the application of the spray treatments, the plants for testing were cutting
Line 311: It is not indicated that chlorophyll was determined in fresh leaves, this can be emphasized for the sake of clarity
Line 315: To be given after how many days of the experiment (from sowing or germination) plants were cutting
Reviewer 2 Report
In this manuscript, the author studied that Foliar spraying with potassium bicarbonate reduces the negative impact of drought stress on sweet basil. The present study aimed to evaluate the effect of potassium bicarbonate (KHCO3) on morphophysiological parameters, phenolics content, and the antioxidant activity of basil under drought conditions. The application of potassium bicarbonate to drought-stressed plants significantly increased chlorophylls content, fresh and dry weight, phenolics content in the two tested cultivars, and antioxidant activity. Principal component analysis showed that the first factor was highly and positively related to all investigated parameters. Hierarchical clustering analysis showed that the first cluster was well-watered, well-watered, and sprayed with potassium bicarbonate and grown under drought conditions and sprayed with potassium bicarbonate basil cultivars—the second cluster – by all tested cultivars grown under drought conditions.
Manuscripts need Serious English editing. It lacks refinements. Also, It did not have proper experiments to indicate drought stress such as leaf wilting, morphological evidence just biochemical analysis would not do justice to this study. Also, the manuscript is heavily plagiarized especially in the discussion part at L234-235, L242-L252, L256-568, L279-284, L288-295, L320-328, L331-333, L344-348, L356-359, L361-365.
Reviewer 3 Report
What is the goal of your work, and what suggestions do you have? Should we focus our efforts on using KHCO3 in drought situations to improve crop yields or to alleviate drought stress?
Abstract: It is not evident from the abstract why this investigation was needed. Why did the author use potassium bicarbonate? The reason to choose potassium bicarbonate as these basil cultivars?
Treatment conditions of experiment must be mentioned in the abstract and was this study repeated or not?
Please rewrite your abstract in a better way, including key words.
Introduction: The reasons for choosing potassium carbonate salt are not evident from the abstract as well as from the introduction.
The aims and objectives of this paper are unclear. There should be a strong hypothesis on which this study was based, which is missing in this paper.
Authors are advised to add a para explaining how KHCO3 increases the growth and other parameters by decreasing drought stress. Potassium is associated with nutrients as it’s classified as a macronutrient. Do authors give some suggestions about it?
The last paragraph of the introduction must be about your purpose of study. What exactly is it that you'll argue for?
Results: In line 72, it should be chlorophyll instead of chlorophylls.
I think it’s better to write total phenolic content (TPC) instead of total phenolics content. Check lines 123, 125, 127, and 130, as well as throughout the manuscript.
The author should be interrupted and discuss all the results in sequence. After figure.2 author is unfolding the fresh weight of plants to remove it (lines 91–93).
Avoid sentence repetition; see lines 91-93 and 105-107 for examples.
The 3 of KHCO3 must be superscript throughout the manuscript, especially in figures.
Discussion: Chlorophyll activity and photosynthetic activity are the same thing (line 214).
In line 216, I think authors should write "significant reduced" instead of "highest decreasing" and in line 218 "decreased" instead of decreasing
I think chlorophyll is a better term than chlorophyll.
Avoid long, convoluted sentences in your discussion. Check line 215-221. The sentence is ambiguous. Make it clear.
Check line 226-227 where the line ends.
From where did line 227 start? Remove "in studies" at the end of the sentence (236-238).
What could be the possible reason for the high root and shoot dry mass in KHCO3 treated plants?
The manuscript should contain recent studies
There are punctuation mistakes and idiosyncrasies in grammar throughout the manuscript.
Material and methods: According to authors “The seedlings with the 5–6 true leaves were divided into 2 groups and one group of seedlings were sprayed with 10 mM potassium bicarbonate (KHCO3). Three days after potassium bicarbonate treatment half of the plants from each group were subjected to water deficit (drought stress) and the other half were grown under stress-free conditions (well- watered) (line 304-307)”. Can author explain this because figures in the results are presenting something else. Was KHCO3 also given to the plants in the drought condition? Because according to the authors treatment has already been given before dividing into groups.
Statistical analysis in material and methodology should be described better.
The conclusion is missing. The authors need to write their suggestions and recommendations as part of the conclusion.